# FedScale: Benchmarking Model and System Performance of Federated Learning

Fan Lai, Yinwei Dai, Xiangfeng Zhu, Mosharaf Chowdhury

*University of Michigan*

## Abstract

We present FedScale, a diverse set of challenging and realistic benchmark datasets to facilitate scalable, comprehensive, and reproducible federated learning (FL) research. FedScale datasets are large-scale, encompassing a diverse range of important FL tasks, such as image classification, object detection, language modeling, speech recognition, and reinforcement learning. For each dataset, we provide a unified evaluation protocol using realistic data splits and evaluation metrics. To meet the pressing need for reproducing realistic FL at scale, we have also built an efficient evaluation platform to simplify and standardize the process of FL experimental setup and model evaluation. Our evaluation platform provides flexible APIs to implement new FL algorithms and includes new execution backends with minimal developer efforts. Finally, we perform indepth benchmark experiments on these datasets. Our experiments suggest fruitful opportunities in heterogeneity-aware co-optimizations of the system and statistical efficiency under realistic FL characteristics. FedScale is open-source with permissive licenses and actively maintained,[1] and we welcome feedback and contributions from the community.

## 1 Introduction

Federated learning (FL) is an emerging machine learning (ML) setting where a logically centralized coordinator orchestrates many distributed clients (e.g., smartphones or laptops) to collaboratively train or evaluate a model [14, 32] (Figure 1). It enables model training and evaluation on end-user data, while circumventing high cost and privacy risks in gathering the raw data from clients, with applications in diverse domains: for example, NVIDIA applies FL to create medical imaging AI [38]; Google runs federated training of NLP models in Google keyboard [17, 55];

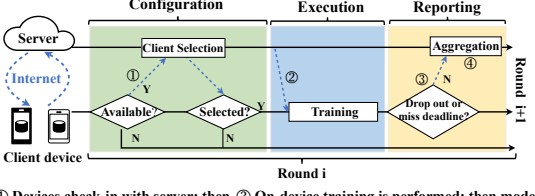

① Devices check-in with server; then sever selects a subset of clients

② Model and configuration are sent to selected devices

③ On-device training is performed; then model update is reported back if training succeeds

④ Server aggregates updates into the global model; then training moves to next round

Figure 1: Standard FL protocol [14, 54].

Apple performs federated evaluation and tuning of automatic speech recognition models on end-user devices [43]; IBM is deploying FL infrastructure to help detect financial misconducts [39].

To address challenges arising from the heterogeneous execution speeds of client devices as well as non-IID data distributions, existing efforts have focused on optimizing different aspects of FL: (1) *System efficiency*: reducing computation load (e.g., using smaller models [47]) or communication traffic (e.g., local SGD [42]) to achieve faster on-device execution; (2) *Statistical efficiency*: designing

---

[1]FedScale is available at https://github.com/SymbioticLab/FedScale.

Submitted to the 35th Conference on Neural Information Processing Systems (NeurIPS 2021) Track on Datasets and Benchmarks. Do not distribute.

| Features | OARF [30] | LEAF [15] | FedEval [16] | FedML [28] | Flower [13] | **FedScale** |
|---|---|---|---|---|---|---|
| Heter. Client Dataset | ◯ | ◯ | ✗ | ◯ | ◯ | ✔ |
| Heter. System Speed | ✗ | ✗ | ✗ | ✗ | ✗ | ✔ |
| Client Availability | ✗ | ✗ | ✗ | ✗ | ✗ | ✔ |
| Scalable Platform | ✗ | ✗ | ◯ | ◯ | ✔ | ✔ |
| Real FL Runtime | ✗ | ✗ | ✗ | ✗ | ✗ | ✔ |
| Flexible APIs | ✗ | ✗ | ✗ | ✔ | ✔ | ✔ |

Table 1: Comparing FedScale with existing FL benchmarks and libraries. ◯ implies limited support. We provide more details for this comparison in Appendix B.

data heterogeneity-aware algorithms (e.g., client clustering [26]) to obtain better training accuracy with fewer training rounds; (3) *Privacy and security*: developing reliable strategies (e.g., differentially private training [31]) to make FL more privacy-preserving and robust to potential attacks.

The performance of an FL solution greatly depends on the characteristics of data, device capabilities, and participation of clients; overlooking any one aspect can mislead FL evaluation (§2). For example, dynamics of client system performance or availability (e.g., device drop-out or rejoining) can affect the dynamics of data availability (distribution shift of cross-device data), which may impair model convergence [20]; too few clients can lead to unstable statistical training convergence, but too many can slow down practical model aggregation because of heterogeneous system speed. As such, a comprehensive suite of benchmarks that combine diverse aspects of practical FL is crucial for systemic evaluation and comparison of different efforts.

Existing benchmarks for FL are mostly borrowed from traditional ML benchmarks (e.g., MLPerf [40]) or designed for simulated FL environments like TensorFlow Federated [12] or PySyft [8]. As shown in Table 1, existing benchmarks for FL fall short in multiple ways: (1) they are limited in the versatility of data for various real-world FL applications. Indeed, even though they may have quite a few datasets and FL training tasks (e.g., FedEval [16] and LEAF [15]), their datasets often contain synthetically generated partitions derived from conventional datasets (e.g., CIFAR) and do not represent realistic characteristics; (2) existing benchmarks often overlook different aspects of practical FL. For example, system speed and availability of the client are largely missing (e.g., FedML [8] and Flower [13]), which discourages FL efforts from considering system efficiency and leads to overly optimistic statistical performance (§2); (3) their experimental environments are unable to reproduce the practical scale of FL deployments. While real FL often involves thousands of participants in each training round [32, 55], existing benchmarking platforms – therefore, many existing FL solutions – are merely able to support the training of tens of participants per round; (4) they may lack user-friendly APIs for automated integration, resulting in great engineering efforts in benchmarking new plugins.

**Contributions:** In this paper, we introduce FedScale, an FL benchmark to empower comprehensive and standardized FL evaluations. As shown in Figure 2, we make the following contributions:

- To the best of our knowledge, we incorporate the most comprehensive FL datasets for evaluating different aspects of real FL deployments. FedScale currently has 18 realistic FL datasets spanning across small, medium, and large scales for a wide variety of task categories, such as image classification, object detection, language modeling, speech recognition, machine translation, recommendation, and reinforcement learning. To account for practical client behaviors, we include real-world measurements of mobile devices, and associate each client with his computation and communication speeds, as well as availability status dynamics.

- We build an automated evaluation platform, FedScale Automated Runtime (FAR), to simplify and standardize the FL evaluation in a more realistic setting. FAR integrates real FL statistical and system dataset, and

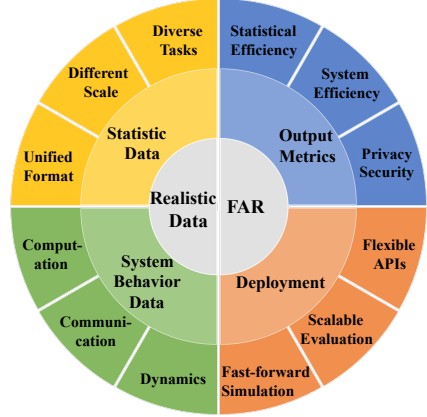

Figure 2: FedScale provides real FL data and an automated evaluation platform.

thus can pinpoint various practical FL metrics needed in today's work. FAR allows easy deployment of new plugins with flexible APIs and can perform the training of thousands of clients in each round on a few GPUs efficiently. FAR is built atop of our recent work Oort [36], which has passed a rigorous artifact evaluation in OSDI 2021.

- We perform indepth benchmark experiments for recent FL efforts in FedScale setting, and highlight the pressing need of co-optimizing system and statistical efficiency in a heterogeneity-aware manner, especially in tackling system stragglers and biased model performance.

## 2 Background

**Existing efforts optimize for various goals of practical FL**   To tackle heterogeneous client data, FedProx [37], FedYogi [44] and Scaffold [33] introduce adaptive client/server optimizations that use control variates to correct for the 'drift' in model updates. Instead of training a single global model, some efforts resort to training a mixture of models [19, 22], clustering clients over training [27], or enforcing guided client selection [36]; To tackle the scarce and heterogeneous device resource, FedAvg [42] reduces communication cost by performing multiple local SGD steps, while some works compress the model update by filtering out or quantizing unimportant parameters [46, 34]; After realizing the privacy risk in FL [24, 51], DP-SGD [25] enhances the privacy by introducing differential privacy, and DP-FTRL [31] applies the tree aggregation to add noise to the sum of mini-batch gradients to ensure privacy further. These FL efforts often navigate privacy-accuracy-computation trade-offs. As such, a realistic FL setting is crucial for comprehensive evaluations.

**Existing FL benchmarks can be misleading**   Existing benchmarks often lack realistic client statistical and system behavior datasets, and/or fail to reproduce large-scale FL deployments.
Unfortunately, these limitations imply that they are not only insufficient for benchmarking diverse FL optimizations, but they can even mislead performance evaluations: (1) As shown in Figure 3(a), the statistical performance becomes worse when encountering practical client behaviors (e.g., stragglers and training failures), which indicates that existing benchmarks that do not have systems traces can produce overly optimistic statistical performance by overlooking systems characteristics; (2) FL training with hundreds of participants each round performs better than that with tens of participants (Figure 3(b)). As such, existing benchmark platforms can under-report existing FL optimizations as they cannot support the practical FL scale with a large number of participants.

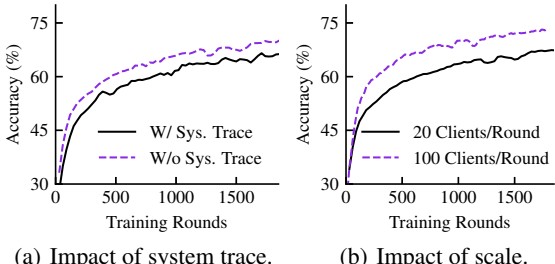

(a) Impact of system trace.   (b) Impact of scale.

Figure 3: Existing benchmarks can mislead.[2]

## 3 FedScale Dataset: Realistic Workloads for Federated Learning

FL performance relies on at least three aspects: (1) *Client statistical data*: the client dataset used for training or testing determines the statistical efficiency of FL tasks (e.g., convergence and model accuracy); (2) *Client system behavior*: the compute/communication speed of the client device and its availability over time determine the system efficiency of FL tasks (e.g., duration of each round and physical cost) and the availability of statistical data; and (3) *Task categories*: model and application combinations that are running can exhibit different reliance on client statistical data and execute at different system speeds. Because client data is tightly coupled with the client device, these aspects interplay with each other and can impact the performance of an FL optimization, be it for statistical efficiency, system efficiency, or privacy. As such, an ideal suite of FL benchmarking dataset should cover all three aspects and support FL deployments at diverse scales.

We next introduce how we collected and partitioned realistic datasets in order to generate a versatile suite of FL datasets provided in FedScale.

---

[2]We train the ShuffleNet model on OpenImage classification task. More experimental setups in Section 5.

| Category | Name | Data Type | #Clients | #Instances | Example Task |
|----------|------|-----------|----------|------------|--------------|
| **CV** | iNature | Image | 2,295 | 193K | Classification |
| | OpenImage | Image | 13,771 | 1.3M | Classification, Object detection |
| | Google Landmark | Image | 43,484 | 3.6M | Classification |
| | Charades | Video | 266 | 10K | Action recognition |
| | VLOG | Video | 4,900 | 9.6K | Classification, Object detection |
| | Waymo Motion | Video | 496,358 | 32.5M | Motion prediction |
| **NLP** | Europarl | Text | 27,835 | 1.2M | Text translation |
| | Blog Corpus | Text | 19,320 | 137M | Word prediction |
| | Reddit | Text | 1,660,820 | 351M | Word prediction |
| | CoQA | Text | 7,189 | 114K | Question Answering |
| | LibriTTS | Text | 2,456 | 37K | Text to speech |
| | Google Speech | Audio | 2,618 | 105K | Speech recognition |
| | Common Voice | Audio | 12,976 | 1.1M | Speech recognition |
| **Misc ML** | Taobao | Text | 182,806 | 20.9M | Recommendation |
| | Fox Go | Text | 150,333 | 4.9M | Reinforcement learning |

Table 2: Statistics of *partial* FedScale datasets (the full list and more details of data and its partition are in Appendix A). FedScale has 18 real-world federated datasets; each dataset is partitioned by its real client-data mapping. Note that we remove the sensitive information in these datasets.

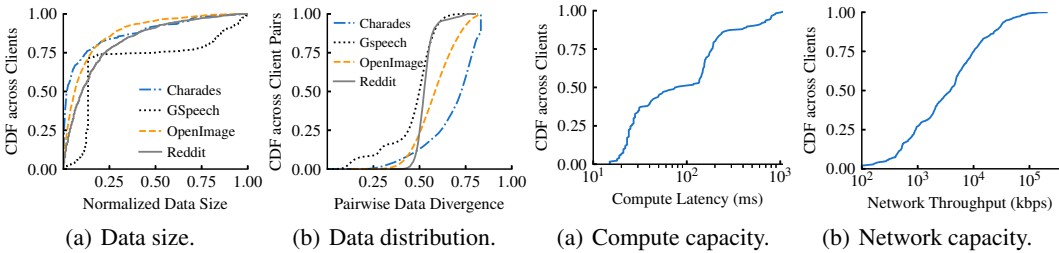

(a) Data size.    (b) Data distribution.    (a) Compute capacity.    (b) Network capacity.

Figure 4: Non-IID client data distribution.      Figure 5: Heterogeneous client system speed.

## 3.1 Client Statistical Dataset

FedScale currently has 18 realistic FL datasets (Table 2), which can be used in various FL tasks (e.g., federated training/testing or on-device fine-tuning). The raw data of these datasets are collected from different sources and stored in various formats. We clean up the raw data, partition them into new FL datasets, and streamline new datasets into consistent formats. Moreover, we categorize them into different FL use cases and provide Python APIs for integrating them into today's frameworks.

**Realistic data and partitions** We target realistic datasets with client information, and partition the raw dataset using the unique client identification. For example, OpenImage is a vision dataset collected by Flickr, wherein different mobile users upload their images to the cloud for public use. We use the AuthorProfileUrl attribute of the OpenImage data to map data instances to each client, whereby we extract the realistic distribution of the raw data. Following existing FL deployments [55], for each dataset, we assign its clients into the training, validation or testing groups, whereby we get the training, validation and testing set for it. Here, we pick four real-world datasets – video (Charades), audio (Google Speech), image (OpenImage), and text (Reddit) – to illustrate the characteristics of FL. Each dataset consists of hundreds or up to millions of clients and millions of data points. Figure 4 reports the *Cumulative Distribution Function* (CDF) of the data distribution, wherein we see a high statistical deviation across clients not only in the quantity of samples (Figure 4(a)) but also in the data distribution (Figure 4(b)).[3] Our findings confirm the non-IID data distribution in FL.

---

[3]We report the pairwise Jensen–Shannon distance of the categorical distribution between two clients.

**Different scales across diverse task categories**   To accommodate diverse scenarios in practical FL, FedScale includes small-, medium-, and large-scale datasets across a wide range of tasks, from hundreds to millions of clients. Some datasets can be applied in different tasks, as we enrich their use case by driving different metadata from the same raw data. For example, the raw `OpenImage` dataset can be used for object detection, and we extract each object therein and generate a new dataset for image classification. Moreover, we provide APIs for the developer to customize their dataset (e.g., enforcing new data distribution or extracting a subset of clients).

## 3.2   Client System Behavior Dataset

**Client device system speed is heterogeneous**   We formulate the system trace of different clients using *AI Benchmark* [1] and *MobiPerf Measurements* [7] on mobiles. *AI Benchmark provides the training and inference speed of diverse models (e.g., MobileNet) across a wide range of device models (e.g., Huawei P40 and Samsung Galaxy S20), while MobiPerf has collected the available cloud-to-edge network throughput of over 100k world-wide mobile clients.*  As specified in real FL deployments [14, 55], we focus on mobile devices that have larger than 2GB RAM and connect with WiFi; Figure 5 reports that their compute and network capacity can exhibit order-of-magnitude difference. As such, how to orchestrate scarce resources and mitigate stragglers are paramount for high system efficiency.

**Client device availability is dynamic**   We incorporate a large-scale user behavior dataset spanning 136k users [54] to emulate the behaviors of clients. It includes 180 million trace items of client devices (e.g., battery charge or screen lock) over a week. We follow the real FL setting, which considers the device in charging to be available [12] and observe great dynamics in their availability: (i) the number of available clients reports diurnal variation (Figure 6(a)). This confirms the cyclic patterns in the client data, which can deteriorate the statistical performance of FL [20]. (ii) the duration of each available slot is not long-lasting (Figure 6(b)).

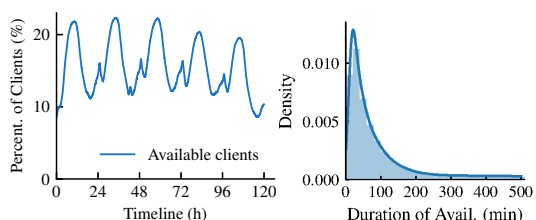

(a) Inter-device availability. (b) Intra-device availability.

Figure 6: Client availability is dynamic.

This highlights the need of handling failures (clients become offline) during training, since the duration of each round (also a number of minutes) is comparable to that of each available slot.

# 4   FAR: Evaluation Platform for Federated Learning

Existing FL evaluation platforms can hardly reproduce the scale of practical FL deployments and fall short in providing user-friendly APIs, which requires great developer efforts to deploy new plugins. As such, we introduce FedScale Automated Runtime (FAR), an automated and easily-deployable evaluation platform, to simplify and standardize the FL evaluation under a practical setting.  FAR is based on our Oort project [36], which has been shown to scale well and can emulate FL training of thousands of clients in each round.

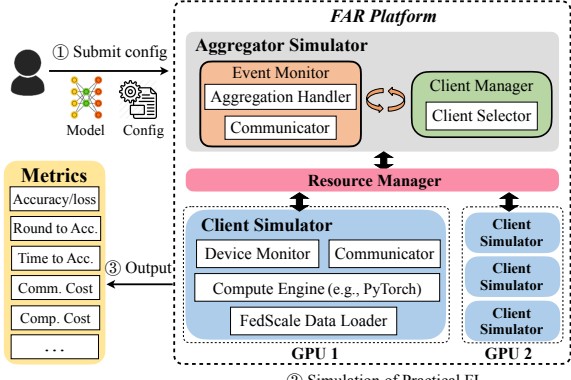

Figure 7: FAR enables the developer to benchmark various FL efforts with practical FL data and metrics.

**Overview of FedScale Automated Runtime (FAR)**   FAR is an automated evaluation platform that can emulate realistic FL behaviors on GPU/CPU, while providing

| Module | API Name | Example Use Case |
|---|---|---|
| **Aggregator Simulator** | `round_initialization_handler`(*args)
`round_completion_handler`(*args)
`client_completion_handler`(client_id, msg)
`push_msg_to_client`(client_id, msg) | Client clustering
Adaptive/secure model aggregation
Straggler mitigation
Model compression |
| **Client Manager** | `select_clients`(*args)
`select_model_for_client`(client_id) | Client selection
Adaptive model selection |
| **Client Simulator** | `train`(client_data, model, config)
`push_msg_to_aggregator`(msg) | Local SGD/malicious attack
Model compression |

Table 3: Some example APIs. FedScale provides APIs to deploy new plugins for various designs.

various practical FL metrics, such as computation/communication cost, latency and wall clock time, for evaluating today's efforts. As shown in Figure 7, FAR primarily consists of three components:

- *Aggregator Simulator*: It acts as the aggregator in practical FL, which selects participants, distributes execution profiles (e.g., model weight), and handles result (e.g., model updates) aggregation. In each round, its client manager uses the client behavior trace to decide whether a client is available; then it selects the specified number of clients to participate that round. Once receiving new events, the event monitor will activate the handler (e.g., aggregation handler to perform model aggregation), or the communicator to send/receive messages. The communicator records the size (cost) of every network traffic, and its FL runtime latency ($\frac{traffic\_size}{client\_bandwidth}$).

- *Client Simulator*: It works as the client in FL. FedScale data loader loads the federated dataset of that client and feeds this data to the compute engine to run real training/testing. The computation latency is determined by ($\#\_processed\_sample \times latency\_per\_sample$), and the communicator handles the network traffics and records the communication latency ($\frac{traffic\_size}{client\_bandwidth}$). At the same time, the device monitor handles different function calls specified by the developer; it will also terminate the simulation of this client and report failure(s) if the current runtime exceeds the available slot (indicated in the client availability trace).

- *Resource Manager*: It orchestrates the available physical resource for evaluation to maximize the utilization of resource. For example, when the number of participants in that round exceeds the resource capacity (e.g., simulating thousands of clients on a few GPUs), the resource manager queues the overcommitted tasks of clients and schedules a new client simulation request from this queue once resource becomes available.

Note that capturing runtime performance (e.g., wall clock time of training) is rather slow in practical FL (each client takes several minutes), but FAR enables *fast-forward* simulation for interactive development, since the real training on our platform often takes only a few seconds per round.

**FAR enables automated and standardized FL simulation** FAR incorporates realistic FL traces, using the aforementioned trace by default, to automatically emulate the practical FL workflow: ① *Task submission*: FL developers specify their configurations (e.g., model and dataset), which can be federated training or testing, and the FAR resource manager will initiate the aggregator and client simulator on available resource (GPU, CPU, other accelerators, or even smartphones); ② *FL simulation*: This evaluation stage follows the standardized FL lifecycle (in Figure 1). In each training round, the aggregator inquires the client manager to select the participants, whereby the resource manager distributes the client configuration to the available client simulators. After the completion of each client, the client simulator pushes the model update to the aggregator, which then performs the model aggregation. ③ *Metrics output*: During training, the developer can query the practical evaluation metrics on the fly. Figure 7 lists some popular metrics supported in FAR.

**FAR is easily-deployable and extensible for plugins** FAR provides flexible APIs, which can accommodate with different execution backends (e.g., PyTorch and TensorFlow) by design, for the developer to quickly deploy new plugins for customized evaluations. Table 3 illustrates some example APIs that can facilitate diverse FL efforts, and Figure 9 dictates an example showing how these APIs

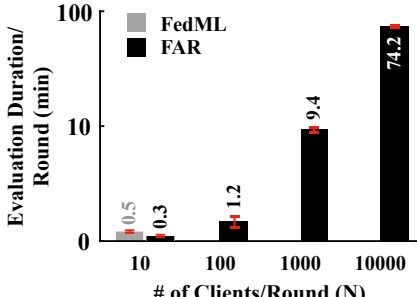

Figure 8: FAR can support thousands of clients per round, while FedML failed to run even 100 clients.

```python
from fedscale.core.client import Client

class Customized_Client(Client):
# Redefine training (e.g., for local
    SGD/gradient compression)
  def train(self,client_data,model,conf):
      # Code of plugin
          ...
      # Results will be sent to aggregator
      return training_result
```

Figure 9: Add plugins by inheritance.

help to benchmark a new design of local client training with a few lines of code. Specifically, the developer can redefine client training function `run_client` by inheriting the base Executor module, and this plugin will be automatically integrated into FedScale during evaluations. Moreover, FAR can embrace new realistic (statistical client or system behavior) datasets with the built-in APIs. For example, the developer can import his own dataset of the client availability by leveraging the API (`load_client_availability`), and FAR will automatically force this trace during evaluations. We also provide more examples in Appendix C to demonstrate the ease of evaluating different today's FL algorithms in FAR– a few lines are all we need!

**FAR is scalable and efficient**   FAR can perform large-scale simulations (e.g., thousands of participants in each round) in both standalone (single CPU/GPU) and distributed (multiple CPUs/GPUs) setting. This is because: (1) FAR can support multiprocessing on a single GPU so that multiple client simulators can co-locate on the same GPU; (2) our resource manager monitors the fine-grained resource utilization of machines, queues the overcommitted simulation requests, adaptively dispatches simulation requests of the client across machines to achieve load balance, and then orchestrates the simulation based on the client mirror clock; (3) FAR maximizes the resource utilization by overlapping the communication and computation phrases of different clients. For example, the simulator can turn to train new clients while the communication of the last client is on the fly. As shown in Figure 8 [4], FAR not only runs faster than FedML [28] (using 10 clients per round), thus saving lots of GPU hours, but can support large-scale evaluations efficiently. Instead, state-of-the-art platforms hardly support the practical FL scale with hundreds of clients, because they mostly rely on the traditional ML architectures (e.g., the primitive parameter-server architecture), which are primarily designed for the traditional ML training on a number of workers with large batch size.

## 5   Experiments

In this section, we first show how FedScale can benefit the benchmarking of existing efforts optimizing for different aspects of FL. Moreover, we highlight some important insights to improve practical FL.

**Experimental setup**   We use 10 NVIDIA Tesla P100 GPUs in our evaluations. Following the real FL deployments [14, 55], the aggregator collects updates from the first $N$ completed participants out of $1.3N$ participants to mitigate system stragglers in each round, and $N = 100$ by default. We pick two representative datasets in FedScale, which belong to different scales and tasks: (1) *Speech Recognition*: the small-scale Google Speech dataset, with 105K speech commands over 2600 clients. We train ResNet-18 [29] to recognize the command among 35 categories. (2) *Image Classification*: the middle-scale OpenImage dataset, with 1.3M images spanning 600 categories across 14k clients. We train ShuffleNet-V2 [57] to classify the image. These applications and models are widely used on mobile devices. We set the minibatch size of each participant to 20, and the number of local steps $K$ to 20. We cherry-pick the hyper-parameters with grid search, ending up with an initial learning rate 0.04. These settings are consistent with the literature.

---

[4]We train the ShuffleNet model on OpenImage classification task. More experimental setups in Section 5.

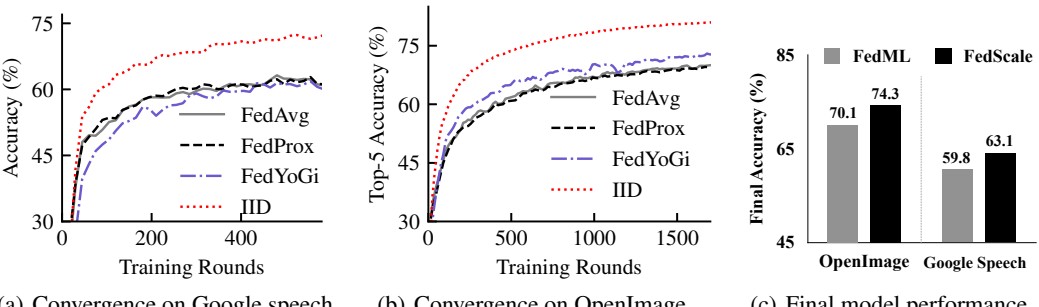

(a) Convergence on Google speech.  (b) Convergence on OpenImage.  (c) Final model performance.

Figure 10: FedScale can benchmark the statistical FL performance. (c) shows existing benchmarks can under-report the FedYoGi performance as they cannot support a large number of participants.

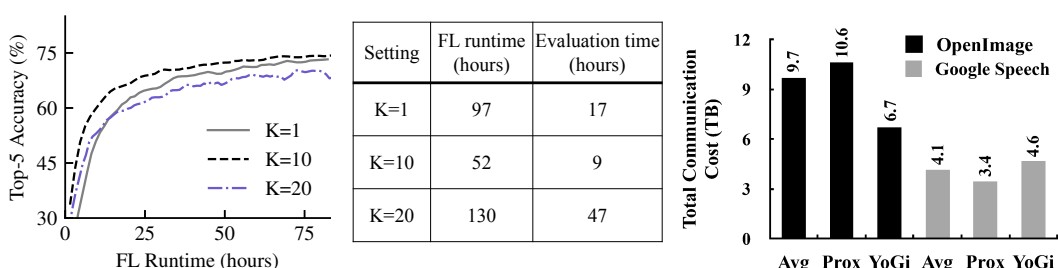

(a) FAR reports realistic FL clock. (b) FAR enables fast-forward eval. (c) FAR reports FL communication cost.

Figure 11: FedScale can benchmark realistic FL runtime. (a) and (b) report FedYoGi results on OpenImage with different number of local steps (K); (b) reports the FL runtime to reach convergence.

### 5.1   How Does FedScale Help FL Benchmarking?

Existing benchmarks are insufficient to evaluate the various metrics needed in today's FL, and can even mis-report the FL performance due to their inability to reproduce the FL setting. Next, we crystallize the effectiveness of FedScale in benchmarking the different FL aspects over its counterparts.

**Benchmarking FL statistical efficiency.**   FedScale provides various realistic client datasets to benchmark the statistical efficiency of FL optimizations. Here, we experiment with three state-of-the-art optimizations (FedAvg, FedProx and FedYoGi) – each reinvents local SGD to mitigate the data heterogeneity – and the traditional IID data setting. Figure 10 reports the statistical training convergence, and we observe that: (1) while the round-to-accuracy performance and final model accuracy of non-IID settings are consistently worse than that of the IID setting, different tasks can have different preferences on the optimizations. For example, FedYoGi performs the best on OpenImage, but it is inferior to FedAvg on Google Speech. Existing benchmarks, however, are limited to quite a few FL tasks and scales, which can discourage the evaluation of FL efforts; and (2) existing benchmarks can under-report the FL performance due to their inability to reproduce the FL setting. Figure 10(c) reports the final model accuracy using FedML and FedScale, where we attempt to reproduce the scale of practical FL with 100 participants per round in both frameworks, but FedML can only support 30 participants because of its suboptimal scalability. We notice this inability of existing benchmarks caps the practical FL performance that the algorithm can indeed achieve.

**Benchmarking FL system efficiency.**   Existing system optimizations for FL focus on the practical runtime (e.g., wall-clock time in real FL training) and the FL execution cost. Unfortunately, existing benchmarks can hardly evaluate the FL runtime due to the lack of realistic system traces, but we now show how FedScale can help such benchmarking: (1) FAR enables fast-forward evaluations of the practical FL wall-clock time with fewer evaluation hours. Taking different number of local steps $K$ in local SGD as an example [42], Figure 11(a) and Table 11(b) illustrate that FedScale can evaluate this

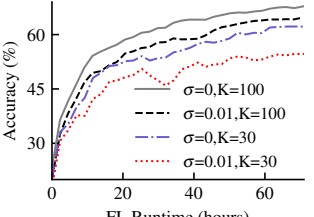

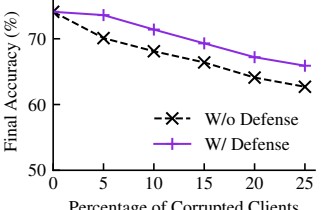

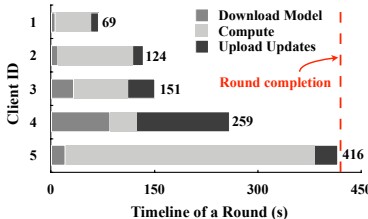

Figure 12: FedScale can benchmark privacy efforts in more realistic FL settings.

Figure 13: FedScale can benchmark security optimizations with realistic FL data.

Figure 14: System stragglers greatly slow down model aggregation in practical FL.

impact of $K$ on practical FL runtime in a few hours. This allows the developer to evaluate large-scale system optimizations efficiently; and (2) FAR can dictate the FL execution cost by using realistic system traces. For example, Figure 11(c) reports the practical FL communication cost in achieving the performance of Figure 10, while Figure 14 reports the system duration of individual clients. These system metrics can facilitate developers to navigate the accuracy-cost trade-off.

**Benchmarking FL privacy and security.** FedScale can evaluate the statistical and system efficiency for privacy and security optimizations in more realistic FL settings than its counterparts. Here, we give an example of how FedScale can benchmark the DP-SGD [25, 31], which applies differential privacy to improve the client privacy. We experiment with different privacy target $\sigma$ ($\sigma$=0 indicates no privacy enhancement) and different number of participants per round $K$. Figure 12 shows that the current scale of participants (e.g., $K$=30) that today's benchmarks can support can mislead the privacy evaluations too: while we notice great performance degradation in the training convergence of taking the privacy optimization (i.e., $\sigma$=0.01) when $K$=30, this performance drop is decent in the practical FL scale ($K$=100). Instead, FedScale is able to benchmark their performance in more FL realistic settings for various privacy use cases, such as wall-clock time, communication cost introduced in the privacy optimization, and the number of rounds needed to leak the privacy on realistic client data.

As for benchmarking the FL security, we follow the example setting of recent backdoor attacks [50, 51] on the OpenImage, where corrupted clients flip their ground-truth labels to poison the training. We benchmarked two settings: one without security enhancement, while the other one clips the model updates as [50]. As shown in Figure 13, state-of-the-art optimizations can mitigate the attacks without hurting the overall performance when a small fraction of clients are corrupted. However, more enhancements are needed as we notice a great accuracy drop as more clients become corrupted.

## 5.2 Opportunities for Future FL Optimizations

**Heterogeneity-aware co-optimizations of communication and computation** Existing optimizations for the system efficiency often apply the same strategy on all clients (e.g., using the same number of local steps [42] or compression threshold [46]), while ignoring the heterogeneous client system speed. When we outline the timeline of 5 randomly picked participants in our training of the ShuffleNet (Figure 14), we find that: (1) system stragglers can greatly slow down the round aggregation in practical FL; and (2) simply optimizing the communication or computation efficiency may not lead to faster rounds, as the last participant can be bottlenecked by the other resource. Here, optimizing the communication can greatly benefit *Client 4*, but it achieves marginal improvement on the round duration as *Client 5* is bottlenecked by computation. As such, there is an urgent need of co-optimizing the communication and computation efficiency while being heterogeneity-aware.

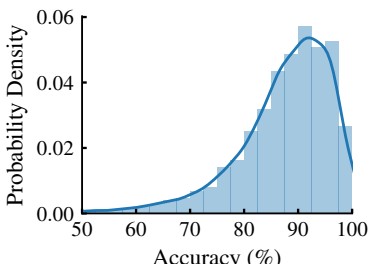

Figure 15: Biased accuracy distributions of the trained ShuffleNet model across clients.

**Co-optimizations of statistical and system efficiency** Most of today's FL efforts focus on either optimizing the statistical or the system efficiency, whereas we observe there exists a great need for jointly optimizing both efficiency: (1) practical FL suf-

fers biased model performance across clients (Figure 15). This can originate from the heterogeneous data and system behaviors, because the system behavior determines the availability of client data over training, wherein predicting this system behavior can curb the statistical drift in advance (e.g., prioritizing the use of upcoming offline clients). Moreover, the popular random client selection can deemphasize clients with slow speed, leading to poor accuracy on slow clients; and (2) statistical optimizations can leverage the heterogeneity nature of client system speed. For example, instead of applying a one-fit-all strategy for all clients, faster workers can trade more system latency against better statistical benefits. For example, faster workers can contribute larger but more accurate model updates when using gradient compression.

## 6   Conclusion

To enable scalable, robust, and reproducible research of federated learning, we introduce FedScale, a diverse set of realistic FL datasets in terms of scales, task categories and client system behaviors. We provide realistic federated datasets for benchmarking today's FL efforts. To enable efficient and standardized FL evaluations, we introduce, FAR, a more scalable evaluation platform than the existing. FAR performs fast-forward evaluation of the practical FL setting and produces FL runtime metrics needed in today's work. More subtly, FAR provides ready-to-use realistic datasets and flexible APIs to allow more FL applications, such as benchmarking the performance of Neural Architecture Search, model inference, and a broader view of federated data analytics (e.g., multi-party computation).

**Societal Impacts and Limitations**   We expect FedScale to be a standard benchmark in federated learning, contributing to the significant advancements of the field. One potential negative impact is that FedScale might narrow down the scope of future papers to the tasks and dataset types that have been included so far. In order to mitigate such a negative impact and limitation, we have made FedScale open-source at: https://github.com/SymbioticLab/FedScale, and will regularly update our datasets and tasks, based on the input from the community.

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

# A   Introduction of FedScale Datasets

| Category | Name | Data Type | #Clients | #Instances | Example Task |
|---|---|---|---|---|---|
| **CV** | iNature [5] | Image | 2,295 | 193K | Classification |
| | FEMNIST [18] | Image | 3,400 | 640K | Classification |
| | OpenImage [4] | Image | 13,771 | 1.3M | Classification, Object detection |
| | Google Landmark [53] | Image | 43,484 | 3.6M | Classification |
| | Charades [49] | Video | 266 | 10K | Action recognition |
| | VLOG [23] | Video | 4,900 | 9.6K | Classification, Object detection |
| | Waymo Motion [21] | Video | 496,358 | 32.5M | Motion prediction |
| **NLP** | Europarl [35] | Text | 27,835 | 1.2M | Text translation |
| | Blog Corpus [48] | Text | 19,320 | 137M | Word prediction |
| | Stackoverflow [10] | Text | 342,477 | 135M | Word prediction, Classification |
| | Reddit [9] | Text | 1,660,820 | 351M | Word prediction |
| | Amazon Review [41] | Text | 1,822,925 | 166M | Classification, Word prediction |
| | CoQA [45] | Text | 7,189 | 114K | Question Answering |
| | LibriTTS [56] | Text | 2,456 | 37K | Text to speech |
| | Google Speech [52] | Audio | 2,618 | 105K | Speech recognition |
| | Common Voice [2] | Audio | 12,976 | 1.1M | Speech recognition |
| **Misc ML** | Taobao [11] | Text | 182,806 | 20.9M | Recommendation |
| | Fox Go [3] | Text | 150,333 | 4.9M | Reinforcement learning |

Table 4: Statistics of FedScale datasets. FedScale has 18 realistic client datasets, which are from the real-world collection, and we partitioned each dataset using its real client-data mapping.

FedScale currently has 18 realistic federated datasets across a wide range of scales and task categories (Table 4). Here, we provide the description of some representative datasets, and the reader can refer to FedScale repository (https://github.com/SymbioticLab/FedScale) for more datasets.

**Google Speech Commands.**   A speech recognition dataset [52] with over ten thousand clips of one-second-long duration. Each clip contains one of the 35 common words (e.g., digits zero to nine, "Yes", "No", "Up", "Down") spoken by thousands of different people.

**OpenImage.**   OpenImage [4] is a vision dataset collected from Flickr, an image and video hosting service. It contains a total of 16M bounding boxes for 600 object classes (e.g., Microwave oven). We clean up the dataset according to the provided indices of clients. In our evaluation, the size of each image is $256 \times 256$.

**Reddit and StackOverflow.**   Reddit [9] (StackOverflow [10]) consists of comments from the Reddit (StackOverflow) website. It has been widely used for language modeling tasks, and we consider each user as a client. In this dataset, we restrict to the 30k most frequently used words, and represent each sentence as a sequence of indices corresponding to these 30k frequently used words.

**VLOG.**   VLOG [23] is a video dataset collected from YouTube. It contains more than 10k Lifestyle Vlogs, videos that people purportedly record to show their lives, from more than 4k actors. This dataset aimed at understanding everyday human interaction and contains labels for scene classification, hand-state prediction task, and hand detection.

**LibriTTS.**   LibriTTS [56] is a large-scale text-to-speech dataset. It is derived from audiobooks that are part of the LibriVox project [6]. There are 585 hours of read English speech from 2456 speakers at 24kHz sampling rate.

**Taobao.**   Taobao Dataset [11] is a dataset of click rate prediction about display Ad, which is displayed on the website of Taobao. It is composed of 1,140,000 users ad display/click logs for

8 days, which are randomly sampled from the website of Taobao. We partitioned it using its real client-data mapping.

**Waymo Motion.**  Waymo Motion [21] is composed of 103,354 segments each containing 20 seconds of object tracks at 10Hz and map data for the area covered by the segment. These segments are further broken into 9 second scenarios (8 seconds of future data and 1 second of history) with 5 second overlap, and we consider each scenario as a client.

## B    Comparison with Existing FL Benchmarks

In this section, we compare FedScale with existing FL benchmarks in more details.

**Data Heterogeneity**    Existing benchmarks for FL are mostly limited in the variety of realistic datasets for real-world FL applications. Even they have various datasets (e.g., LEAF [15]) and FedEval [16]), their datasets are mostly synthetically derived from conventional datasets (e.g., CIFAR) and limited to quite a few FL tasks. These statistical client datasets can not represent realistic characteristics and are inefficient to benchmark various real FL applications. Instead, FedScale provides 18 comprehensive realistic datasets for a wide variety of tasks and across small, medium, and large scales, and these datasets can also be used in data analysis to motivate more FL designs.

**System Heterogeneity**    The practical FL statistical performance also depends on the system heterogeneity (e.g., client system speed and availability of the client), which has inspired lots of optimizations for FL system efficiency. However, existing FL benchmarks have largely overlooked the system behaviors of FL clients, which can produce misleading evaluations, and discourages the benchmarking of system efforts. To emulate the heterogeneous system behaviors in practical FL, FedScale incorporates real-world traces of mobile devices, and associates each client with his system speeds, as well as the availability. Moreover, it is non-trivial to emulate these behaviors at scale, so we develop FAR, which is more efficient than the existing.

**Scalability**    Existing frameworks, perhaps due to the heavy burden of building complicated system support, largely rely on the traditional ML architectures (e.g., the primitive parameter-server architecture of Pytorch). These architectures are primarily designed for the traditional large-batch training on a number of workers, and each worker often trains a single batch at a time. However, this is ill-suited to the simulation of thousands of clients concurrently: (1) they lack tailored system implementations to orchestrate the synchronization and resource scheduling, for which they can easily run into synchronization/memory issues and crash down; (2) their resource can be under-utilized, as FL evaluations often use a much smaller batch size than that in the traditional architecture.

Tackling all these inefficiencies requires domain-specific system designs, and the FAR is refactored atop of our Oort project [36]. Specifically, we first built an advanced resource scheduler: It monitors the fine-grained resource utilization of machines, queues the overcommitted simulation requests, adaptively dispatches simulation requests of the client across machines to achieve load balance, and then orchestrates the simulation based on the client mirror clock. Moreover, given a much smaller batch size in FL, we maximize the resource utilization by overlapping the communication and computation phrases of different client simulations. The former and the latter make FedScale more scalable across machines and on single machines, respectively.

**Modularity**    As shown in Table 1, some existing frameworks (e.g., LEAF and FedEval) do not provide user-friendly modularity, which requires great engineering efforts to benchmark different components, and we recognize that FedML and Flower provide the API modularity in this table too.

On the other hand, FAR's modularity for easy deployments and broader use cases is not limited to APIs (Figure 7): (1) FAR Data Loader: it simplifies and expands the use of realistic datasets. e.g., developers can load and analyze the realistic FL data to motivate new algorithm designs, or imports new datasets/customize data distributions in FedScale evaluations; (2) Client simulator: it emulates the system behaviors of FL clients, and developers can customize their system traces in evaluating the FL system efficiency too; (3) Resource Manager: it hides the system complexity in training large-scale participants simultaneously for the deployment.

```
from fedscale.core.client_manager import ClientManager
import Oort

class Customized_ClientManager(ClientManager):
    def __init__(self, *args):
        super().__init__(*args)
        self.oort_selector = Oort.create_training_selector(*args)

    # Replace default client selection algorithm w/ Oort
    def resampleClients(self, numOfClients, cur_time, feedbacks):
        # Feed Oort w/ execution feedbacks from last training round
        oort_selector.update_client_info(feedbacks)
        selected_clients = oort_selector.select_participants(numOfClients, cur_time)

        return selected_clients
```

Figure 16: Evaluate new client selection algorithm [36].

```
from fedscale.core.client import Client

class Customized_Client(Client):
# Customize the training on each client
  def train(self,client_data,model,conf):
      # Get the training result from
      # the default training component
      training_result = super().train(
          client_data, model, conf)

      # Implementation of compression
      compressed_result = compress_impl(
              training_result)
      return compressed_result
```

```
from fedscale.core.client import Client

class Customized_Client(Client):
# Customize the training on each client
  def train(self,client_data,model,conf):
      # Get the training result from
      # the default training component
      training_result = super().train(
          client_data, model, conf)

      # Clip updates and add noise
      secure_result = secure_impl(
              training_result)
      return secure_result
```

Figure 17: Evaluate model compression [46].     Figure 18: Evaluate security enhancement [50].

## C   Examples of New Plugins

In this section, we demonstrate several examples to show the ease of integrating today's FL efforts for realistic evaluations in FedScale.

At its core, FAR provides flexible APIs on each module so that the developer can access and customize methods of the base class. Note that FAR will automatically integrate new plugins into evaluations, and then produces practical FL metrics. Figure 16 demonstrates that we can easily evaluate new client selection algorithms, Oort [36], by modifying a few lines of the `clientManager` module. Similarly, Figure 17 and Figure 18 show that we can extend the basic `Client` module to apply new gradient compression [46] and enhancement for malicious attack [50], respectively.

## D   FedScale Maintenance

**Availability of data and platform**   We have made FedScale open-source on the Github (https://github.com/SymbioticLab/FedScale). So the code and dataset can be downloaded from this repository. For each dataset, we provide detailed descriptions (`README.md`) of the source, organization, format and use case under the repository. So far, these datasets are host on Dropbox, and we are migrating them to the stable storage of AWS. For the evaluation platform FAR, we provide the configuration and job submission guideline as well. We encourage the reviewer to refer to our repository for more details.

**Maintenance plan and responsibility.**   We are actively updating our benchmark weekly, based on the feedback from the community. Currently, our dataset and platform are subject to the *Apache-2.0*

*License*. We respect the contributor of each dataset in following ways: (1) we provide the scripts for the developer to preprocess the downloaded raw data from its original source. This will absolutely obey the rule of each contributor; (2) for those publicly available and widely-used dataset, we temporally host the processed data on our repository. However, we are creating permissive license for each dataset and acknowledgment to respect their contributor, and highlight all assets in our repository are for research purpose only; (3) for all assets in our work, we have removed the sensitive information and use anonymous information to partition the data; (4) we are keeping in touch with all the contributors, and will fix any issues (e.g., by removing that dataset) once that happens.

