# OpenReview forum: "FedScale: Benchmarking Model and System Performance of Federated Learning"
_NeurIPS.cc/2021/Track/Datasets_and_Benchmarks/Round1 — Submitted to NeurIPS 2021 Datasets and Benchmarks Track (Round 1)_

### Official Review · Reviewer_SNT4 · 2021-06-18
**A very helpful toolkit for benchmarking FL.**

**Rating:** 7
**Confidence:** 3
**Clarity:** This paper is written clearly and ver…

**Strengths:**

Strengths:
- Extensive and realistic datasets: FedScale provides 18 realistically partitioned datasets. The datasets span a wide range of tasks: image classification, translation, object detection, recommendation. Also, the scale of the datasets spans a wide range, from thousands to millions. These datasets would be helpful in evaluating FL algorithms.
- Simulation of systems heterogeneity: Systems heterogeneity is an important problem in FL aside from data heterogeneity. The authors did a very good job in stimulating systems heterogeneity, including the computation/communication performance, and availability.
- Systems implementation: The authors implement FAR, a usable FL API for evaluation. This would be helpful to researchers in FL. The systems is also efficient and implements optimizations, such as communication/computation overlapping.
- Good insights: I appreciate the authors' experiments and insights regarding system & algorithm co-optimization in Section 5.

**Weaknesses:**

- Regarding 'Privacy'. In Section 1 and Fig. 2, the authors claim that the system can evaluate privacy. However, in the paper, the authors only showcase how to perform evaluations on backdoor attacks, which from my opinion is not related to privacy. Privacy means that the individual data should not be revealed, while backdoor attacks do not aim to reconstruct original data, but try to compromise the model trained, which should be more related to 'security' instead of 'privacy'. Therefore, I think that the authors should re-consider their claims in Section 1.

**Additional Feedback:**

Please see the "Weaknesses" above.

Minor issues:
- Table 2 in the main paper contains only 15 datasets, while Table 4 in the appendix contains 18 datasets. Why is the difference?
- Line 37: What is "privacy efficiency"? I cannot understand why this is related to efficiency.
- Line 47: "borrow" -> "borrowed".

Further comments:
- It would be even better if you can try to implement security features into FedScale, e.g. differential privacy. This would help the research community even further.

========================================================

After the update, I think that my concerns have been addressed.

**Correctness:**

The benchmark is generally correct and accurate. I appreciate the authors' efforts in trying to make it realistic, in terms of both data and systems heterogeneity. The only thing that I cannot fully agree with is the notion of 'privacy' as mentioned in the Weaknesses part.

**Documentation:**

From the appendix and the links provided by the authors, I think that there is enough description for the datasets. Compared to data, I think that the documentation on how to use the evaluation APIs is a little insufficient. I would expect a more detailed documentation on the APIs, which would even more facilitate using FAR.

**Ethics:**

No.

**Relation To Prior Work:**

I am somewhat familiar with LEAF and FedEval mentioned in Table 1, and I think that the comparisons with these two benchmarks are correct and clear.

**Summary And Contributions:**

This paper presents an evaluation system for federated learning. The presented system, FedScale, contains multiple datasets and can evaluate both system and statistical performances. The authors also demonstrate that the system is flexible and easily usable.

The contributions of this paper can be summarized as follows:
- Comprehensive datasets: 18 datasets with realistic federated data partitions. The datasets are also diverse, in terms of both scale and tasks.
- Realistic Systems Benchmarking: The paper also incorporates system heterogeneity by simulating computation/communication performance and availability.
- Easily usable runtime: The paper presents FAR, which is a set of APIs to evaluate FL in terms of many metrics.
- Experiments and Insights: The paper does benchmarking experiments and reveal some interesting insights, including system & statistics co-optimization.

---

> ### Author Response · Authors · 2021-07-12
> **Update for comments**
>
> We very much appreciate the positive and constructive feedback. We have updated our draft to rephrase privacy and security, add privacy experiments, and solve other loose ends mentioned. If there remain any concerns, please let us know. We will make our best efforts to address them further.
>
> ***Q1: The authors claim that the system can evaluate privacy. However, in the paper, the authors only showcase how to perform evaluations on backdoor attacks.***
>
> FedScale can help in benchmarking the FL optimizations for privacy too. These optimizations often navigate the privacy-accuracy-system trade-offs in designs: their statistical model performance still relies on the realistic FL client data, while the system efficiency depends on the client system behaviors (e.g., system speed and availability). For example, with FedScale, the developer can benchmark the sensitivity of privacy enhancement in more realistic FL settings: how many rounds (statistical efficiency) needed to leak the privacy [1]? How much communication cost (system cost) will be introduced to reach the same accuracy after applying the privacy optimizations (e.g., differential privacy [2])? How does the training algorithm (e.g., DP-SGD [3]) impact the privacy budget?, and so on.
>
> We added a new experiment (differential privacy) and discussions to show how FedScale can facilitate FL privacy optimizations in our updated version.
>
> [1] Jonas Geiping, Hartmut Bauermeister, Hannah Dröge, and Michael Moeller. Inverting Gradients - How easy is it to break privacy in federated learning?. In arxiv.org/abs/2003.14053, 2020.
>
> [2] Peter Kairouz, Brendan McMahan, Shuang Song, Om Thakkar, Abhradeep Thakurta, and Zheng Xu. Practical and private (deep) learning without sampling or shuffling. In arxiv.org/abs/2103.00039, 2021.
>
> [3] Robin C. Geyer, Tassilo Klein, and Moin Nabi. Differentially private federated learning: A
> client level perspective. In NeurIPS, 2017.

---

### Official Review · Reviewer_FQpP · 2021-07-05
**A step in the right direction for an important problem area**

**Rating:** 6
**Confidence:** 3
**Correctness:** Seems OK.

**Strengths:**

The strongest contribution of the paper lies in the way it accounts for client heterogeneity in its evaluations.

**Weaknesses:**

The simulation / emulation environment based on client traces seems quite primitive. The reviewer wonders if more realistic simulation and emulation environments from computer networking and systems areas could not be taken advantage of here.

The benchmarks by and large ignore privacy, which is the motivating reason for federated learning. While the paper may not contribute much here, it would have been nice to see a discussion.

**Additional Feedback:**

As this paper represents a first step, it would be great to have a discussion of open (unaddressed) issues that need to be addressed in the future.

**Clarity:**

Overall, the paper is well-written. Though it would have been better if the paper also described all the desiderata for such a benchmarking framework, even if FedScale achieves a subset of the desiderata.

**Documentation:**

Yes.

**Ethics:**

The work itself does not raise many ethics concerns. But, it would be nice to see a bit more discussion on whether the existing client traces are representative of client connectivity world-wide and/or whether they are more representative of clients in the developed world.

**Relation To Prior Work:**

Yes.

**Summary And Contributions:**

This paper proposes a set of benchmarking datasets for federated learning. The big novelty with the datasets is that they account for client (end user) heterogeneity in terms of compute & communication resources and availability. Accounting for them, can provide a different perspective on the performance of the various proposed optimizations for federated learning.

---

> ### Author Response · Authors · 2021-07-12
> **Update for comments**
>
> We very much appreciate the positive and constructive feedback. We address the comments as below, however, if there remain any concerns, please let us know. We will make our best efforts to address them further.
>
> ***Q1: Wonder if more realistic simulation and emulation environments from computer networking and systems areas could not be taken advantage of here.***
>
> To the best of our knowledge, this is the state-of-the-art in systems and networking conferences as well. For one example, Oort uses the same methodology and will be presented in the top systems conference, OSDI ‘21, this month. The FedScale backend (FAR) develops from Oort, but with much more improvements: (1) it incorporates more diverse realistic datasets and user-friendly APIs to meet the pressing need of cost-efficient and easily-deployable benchmarking, which enables the community to evaluate various aspects of FL needed in today’s FL work in a more realistic setting than the existing; (2) FedScale is open-sourced, and new implementations can be integrated into the FedScale architecture. For example, we are actively developing new backends (e.g., mobile backends) under FedScale.
>
>
> ***Q2: The benchmarks by and large ignore privacy, which is the motivating reason for federated learning. While the paper may not contribute much here, it would have been nice to see a discussion.***
>
> We recognize that the privacy issue is one primary obstacle that discourages today’s benchmarks in collecting realistic FL datasets for evaluations. To the best of our knowledge, FedScale provides the largest comprehensive suite of FL datasets for the pressing need of benchmarking various aspects of FL performance.
>
> On the other hand, we note that FedScale can help in benchmarking the FL optimizations for privacy too. These optimizations often navigate the privacy-accuracy-system trade-offs in designs: their statistical model performance still relies on the realistic FL client data, while the system efficiency depends on the client system behaviors (e.g., system speed and availability). For example, with FedScale, the developer can benchmark the sensitivity of privacy enhancement in more realistic FL settings: how many rounds (statistical efficiency) needed to leak the privacy [1]? How much communication cost (system cost) will be introduced to reach the same accuracy after applying the privacy optimizations (e.g., differential privacy [2])? How does the training algorithm (e.g., DP-SGD [3]) impact the privacy budget?, and so on.
>
> We added a new experiment and discussions to show how FedScale can facilitate FL privacy optimizations in our updated version.
>
> ***Q3: Whether the existing client traces are representative of client connectivity world-wide?***
>
> Thank you for pointing this out. We have updated our draft to cover this.
>
> [1] Jonas Geiping, Hartmut Bauermeister, Hannah Dröge, and Michael Moeller. Inverting Gradients - How easy is it to break privacy in federated learning?. In arxiv.org/abs/2003.14053, 2020.
>
> [2] Peter Kairouz, Brendan McMahan, Shuang Song, Om Thakkar, Abhradeep Thakurta, and Zheng Xu. Practical and private (deep) learning without sampling or shuffling. In arxiv.org/abs/2103.00039, 2021.
>
> [3] Robin C. Geyer, Tassilo Klein, and Moin Nabi. Differentially private federated learning: A
> client level perspective. In NeurIPS, 2017.

---

### Official Review · Reviewer_TXFp · 2021-07-06
**The paper proposes a new set of datasets and a framework that facilitates practical and scalable evaluation**

**Rating:** 6
**Confidence:** 3

**Strengths:**

The main strength of this paper is the introduction of a realistic and large scale dataset that would bring the community very close to benchmarking FL in the real world.

The authors do a good job in identifying key issues with the existing way of benchmarking FL experiments and provide a solution to some of them. The benchmark experiments that blend resource and statistical efficiency would open up a new direction of methodological FL works that can build upon this benchmark.

The proposed FAR simulator is significantly more scalable than the existing frameworks and hence would be useful for the upcoming FL works that aim at evaluating multiple FL under such setups.

The API design appears to be flexible and modular for ease of integration. The amount of code required to run FL experiments is indeed a few lines of code.


**Weaknesses:**

FAR Platform
- Practicality of the benchmark - Saying that a limited number of clients is not a practical setup is debatable. I would argue that a limited number of clients is a more practical setup when big institutions want to train a model together without centralizing data (due to privacy reasons).

- Modularity - While L233-244 touches upon the framework’s flexibility and modularity, it is not clear why this would be a preferred design over something like FedML that also provides a similar API. I would suggest if this part can be covered somewhat more rigorously (like comparing the architecture or design etc.), it would bolster their claim. Otherwise, it is difficult for me as a reviewer to agree with the claims made. Another good example could have been writing a similar code as shown in Fig. 15, 16, and Fig.17 but using other frameworks.

- Scalability - Point 1) at L247 should not be considered a practical solution because in the practical world different individuals are not sharing the same GPU. However, for resource-efficient simulation, I see it as a good solution. Once again, it would help if you mention what design primitives were missing from the existing works that make FAR better than the others. I am also not entirely sure what authors mean when they say existing frameworks “fail” to support more than 10 clients. Is their framework not able to accommodate more clients due to memory issues or the models are not converging or something else?

Experiments
For the three categories of experiments mentioned from L293-322, it is not clear how this benchmark helps in addressing those problems? In general, it is not clear what value is added by those experiments to the main theme of the paper.
For example - are you proposing a new way of benchmarking or measuring co-optimizations of the system and statistical efficiency? Or is it just that the benchmark can be useful for researchers working on those problems, in such a case it would be useful to mention how existing frameworks are not suitable in comparison to FAR for such problems.
By looking at the third contribution, I assume the authors want it to present these experiments as a comparison between existing works but it is again not clear to me if it adds any value to the main theme.

At a high level, I see two weaknesses -
Lack of comparison with existing works as pointed above
Many claims have been phrased too strongly about the existing works. My suggestion would be for the authors to rewrite such phrases.


**Additional Feedback:**

Once I have the first set of responses from the authors, I believe I can make better suggestions for improvement. For now, I would recommend addressing the items mentioned in the weaknesses.

**Clarity:**

The paper is well written and the flow of the paper is smooth. However, I believe some of the claims can be significantly toned down (mentioned in weaknesses) and should focus on objective results more than subjective evaluations.

**Correctness:**

The dataset construction appears to be practically motivated and coherently presented. I have some concerns with the experiments shown in the paper which I have mentioned in the "Weaknesses" section. However, the evaluation metrics and criteria are appropriate.

**Documentation:**

Since the work uses existing datasets, it is not required to mention the data collection, their contribution (processing) has been discussed in the paper. The maintenance plan is well specified in the supplementary.

**Ethics:**

I don't see any ethical concerns.

**Relation To Prior Work:**

The paper fixes the scalability issue for a multiple clients setup in comparison to existing works such as Flower, FedML, Leaf and FedEval. However, the experiments and comparisons made are not enough from an objective standpoint. However this is justifiable because of heavy compute required to do large scale training for all the frameworks and datasets.

**Summary And Contributions:**

The paper makes the following claims about the existing frameworks and benchmarks -
Limitation with existing benchmarks and datasets
 - Lack of versatility in the data
 - Synthetically generated partitions of conventional datasets
Not practical enough
 - System speed and availability is missing
Not reproducing large scale deployments
 - Lack of user-friendly API for automated integration

This paper fixes those aforementioned problems by making the following contributions -
- Integrate existing datasets in a way which is more suitable for practical FL
- Modular Framework
- Experiments on some of the existing problems/techniques.

---

> ### Author Response · Authors · 2021-07-12
> **Update for comment on FAR design**
>
> We very much appreciate your feedback. We updated our description to tone down the comparison throughout the paper, and added detailed comparisons with existing benchmarks in Appendix B and experiments in Section 5.1. If there remain any concerns, we are happy to make our best efforts to address them further.
>
> ***Q1: Practicality of the benchmark - Saying that a limited number of clients is not a practical setup is debatable. I would argue that a limited number of clients is a more practical setup.***
>
> Our benchmark is in line with the practical scale of FL deployments in big institutions (e.g., Google[1, 2] and Mozilla[3]), where thousands of participants (out of millions of clients) are involved in each round. Indeed, it has been reported that using a limited number of participants per round can lead to suboptimal model convergence because of the data heterogeneity [3], and we have similar observations too ($5.1).
>
> More subtly, we note that FedScale provides realistic datasets across a wide range of scales (small, medium and large) and practical FL tasks. As such, a number of small-scale FedScale datasets (or the developer can take a subset of large datasets with our APIs) can support the setup with a limited number of clients too, and FedScale can provide more efficient fast-forward simulation (Figure 8), more versatile datasets (Table 2), and more useful metrics (e.g., system cost) than the existing work.
>
> ***Q2: Modularity - it is not clear why this would be a preferred design over others with APIs.***
>
> As shown in Table 1, some existing frameworks (e.g., LEAF and FedEval) do not provide user-friendly modularity, which requires great engineering efforts to benchmark different components, and we recognize that FedML and Flower provide the API modularity in this table too.
>
> On the other hand, FedScale’s modularity for easy deployments and broader benchmarking is not limited to APIs (Figure 7): (1) FedScale Data Loader: it simplifies and expands the use of realistic datasets. e.g., developers can load and analyze the realistic FL data to motivate new algorithm designs, or imports new datasets/customize data distributions in FedScale evaluations; (2) Client simulator: it emulates the system behaviors of FL clients, and developers can customize their system traces in evaluating the FL system efficiency too; (3) Resource Manager: it hides the system complexity in training large-scale participants simultaneously for the deployment.
>
> ***Q3: Scalability - It would help if you mention what design primitives were missing from the existing works that make FAR better than the others. Why do existing frameworks “fail” to support more than 10 clients?***
>
> Existing frameworks, perhaps due to the heavy burden of building complicated system support, largely rely on the traditional ML architectures (e.g., the primitive parameter-server architecture of Pytorch). These architectures are primarily designed for the traditional large-batch training on a number of workers, and each worker often trains a single batch at a time. However, this is ill-suited to the simulation of thousands of clients concurrently: (1) they lack tailored system implementations to orchestrate the synchronization and resource scheduling, for which they can easily run into synchronization/memory issues and crash down; (2) their resource can be under-utilized, as FL evaluations often use a much smaller batch size than that in the traditional architecture.
>
> Tackling all these inefficiencies requires domain-specific system designs, and the FAR prototype is refactored atop of our Oort project, which has passed a rigorous artifact evaluation in the top-tier system conference (OSDI 2021). Specifically, we first built an advanced resource scheduler: It monitors the fine-grained resource utilization of machines, queues the overcommitted simulation requests, adaptively dispatches simulation requests of the client across machines to achieve load balance, and then orchestrates the simulation based on the client mirror clock. Moreover, given a much smaller batch size in FL, we maximize the resource utilization by overlapping the communication and computation phrases of different client simulations. The former and the latter make FedScale more scalable across machines and on single machines, respectively.
>
>
> [1] Keith Bonawitz, Hubert Eichner, Wolfgang Grieskamp, Dzmitry Huba, Alex Ingerman, and et al. Towards federated learning at scale: System design. In MLSys, 2019.
>
> [2] Timothy Yang, Galen Andrew, Hubert Eichner, Haicheng Sun, Wei Li, Nicholas Kong, Daniel
> Ramage, and Françoise Beaufays. Applied federated learning: Improving Google keyboard
> query suggestions. In arxiv.org/abs/1812.02903, 2018.
>
> [3] Florian Hartmann, Sunah Suh, Arkadiusz Komarzewski, Tim D. Smith, and Ilana Segall. Federated Learning for Ranking Browser History Suggestions, In arXiv preprint arXiv:1911.11807.

---

> ### Author Response · Authors · 2021-07-12
> **Update for comment on Evaluations**
>
> ***Q4: For experiments mentioned from L293-322, it is not clear how this benchmark helps in addressing those problems?***
>
> We have updated this section ($5.1) to validate the effectiveness of FedScale compared to its counterparts, and key takeaways are:
>
> 1. FedScale can help the researcher to evaluate various aspects (e.g., system efficiency) of their FL work, while the existing benchmark can be insufficient because of their limited datasets;
>
> 2. FedScale can benchmark FL efforts in a more scalable and realistic setting, while existing benchmarks can be misleading because of their inability to reproduce the FL behaviors;

---

### Decision · Program_Chairs · 2021-07-27

**Decision:**

Reject

**Comment:**

In this paper the authors provide a new framework for evaluating federated learning algorithms across a wide variety of datasets and tasks.  Reviewers were generally impressed with the setup, with the primary two pieces of feedback being (a) a clearer discussion of privacy implications and (b) clearer comparison to other federated learning evaluation tools.  In particular for comparison to other evaluation tools, the authors are encouraged to follow reviewer feedback be more precise and qualified in terms of the unique contributions of this approach over prior approaches.  I'd encourage the authors to revise and resubmit.